# Associations of Serum GIP, GLP-1, and DPP-4 with Metabolic and Hormonal Profiles and Tobacco Exposure in Women with Polycystic Ovary Syndrome

**DOI:** 10.3390/ijms26157097

**Published:** 2025-07-23

**Authors:** Anna Bizoń, Julia Borkowska, Grzegorz Franik, Agnieszka Piwowar

**Affiliations:** 1Department of Toxicology, Faculty of Pharmacy, Wroclaw Medical University, 50-556 Wroclaw, Poland; agnieszka.piwowar@umw.edu.pl; 2Students Scientific Society at the Department of Toxicology, Wroclaw Medical University, 50-556 Wroclaw, Poland; 3Department of Endocrinological Gynecology, Medical University of Silesia, 40-752 Katowice, Poland; gfranik@sum.edu.pl

**Keywords:** polycystic ovary syndrome, incretin hormones, disorders

## Abstract

Disorders in glucose metabolism are well-established features of polycystic ovary syndrome (PCOS) and are linked to its clinical severity and phenotypic variability. This study aimed to assess serum concentrations of glucagon-like peptide-1 (GLP-1), glucose-dependent insulinotropic polypeptide (GIP), and dipeptidyl peptidase-4 (DPP-4) and to examine their relationships with glucose and insulin levels, selected sex hormone concentrations, body weight, and exposure to tobacco smoke. Women with PCOS exhibited significantly elevated levels of fasting glucose, insulin, GIP, and GLP-1 compared to controls. Tobacco smoke exposure in women with PCOS was associated with reduced DPP-4 levels, which were approximately two-fold lower in smokers than in non-smokers. A significant negative correlation between DPP-4 and cotinine levels further supported this relationship. Comorbidities such as overweight/obesity or insulin resistance (IR) were also linked to elevated incretin hormone levels. However, no significant age-related trends in incretin levels were identified, despite the known association between age and glucose dysregulation. The notable alterations in incretin hormone profiles in PCOS, along with the consistent patterns of GIP or GLP-1 with metabolic and hormonal parameters, suggest that these hormones may play coordinated regulatory roles in the pathophysiology of PCOS.

## 1. Introduction

Disorders in glucose metabolism are well-recognized features of polycystic ovary syndrome (PCOS) [1]. Women with PCOS exhibit a significantly higher prevalence of hyperinsulinemia, insulin resistance (IR), impaired glucose tolerance, and type 2 diabetes (T2D) compared to the general female population [2]. These metabolic disturbances are correlated with the clinical severity and specific phenotypes of PCOS [3]. Insulin can augment gonadotropin-releasing hormone (GnRH)-stimulated luteinizing hormone (LH) beta gene expression in pituitary gonadotroph cells. Specifically, insulin enhances the effects of GnRH on LHβ mRNA levels through the transcription factor early growth response protein 1 (Egr-1), indicating a direct action of insulin at the pituitary level [4,5]. This mechanism contributes to hyperinsulinemia being a key driver of excessive androgen production in PCOS [6]. Additionally, insulin inhibits the hepatic synthesis of sex hormone-binding globulin (SHBG), a protein responsible for regulating the availability of circulating testosterone. This suppression leads to decreased SHBG levels in the bloodstream and increased free androgen levels [7], which manifest clinically as hirsutism, alopecia, and acne. Hyperinsulinemia and IR are strong predictors of PCOS and related cardiometabolic disorders as well as psychiatric disorders and many others [8,9]. Physiologically, 50–70% of postprandial insulin secretion is mediated by the incretin effect—an enhanced insulin response following oral glucose intake compared to intravenous administration, despite similar blood glucose levels [10]. This effect is primarily driven by two gut-derived hormones: glucagon-like-peptide-1 (GLP-1) and glucose-dependent insulinotropic polypeptide (GIP). Both hormones act through G protein-coupled receptors, stimulating intracellular cyclic adenosine monophosphate production, which elevates cytosolic calcium concentrations in pancreatic β-cells, ultimately triggering insulin release [11]. It was also shown that GLP-1 enhances insulin secretion, improves insulin sensitivity, increases satiety, delays gastric emptying, and promotes weight loss [12]. Whereas GIP significantly contributes to the accumulation and storage of nutrients, acting directly through insulin-like effects and indirectly by promoting insulin secretion. In individuals with obesity, GIP levels are often elevated, which may worsen IR commonly observed in patients with T2D [13]. Studies suggest that dual GIP/GLP-1 receptor agonists provide beneficial metabolic effects in individuals with type 2 diabetes and obesity [14]. The combined action of both incretins can improve glycemic control, promote significant weight loss, and positively influence lipid profiles and blood pressure [15]. Dipeptidyl peptidase-4 (DPP-4) plays a crucial role in regulating the activity of the incretin hormones GIP and GLP-1. DPP-4 is a widely expressed serine peptidase, present both as a membrane-bound protein in tissues such as the liver, intestines, kidney, and vascular endothelium, and in a soluble form in plasma. Its primary function in glucose metabolism is to inactivate incretin hormones by cleaving their N-terminal dipeptides. As a result, both GIP and GLP-1 are rapidly degraded by DPP-4, significantly limiting their biological activity. Pharmacological inhibition of DPP-4 increases the half-life and circulating levels of active incretins, enhancing glucose-dependent insulin secretion and reducing glucagon release—both contributing to improved glycemic control [16]. Lifestyle interventions, including increased physical activity and a balanced diet, can improve metabolic and reproductive outcomes in women with PCOS. However, pharmacological treatments such as metformin or GLP-1 and GIP analogues are often necessary. A meta-analysis by Creanga et al. [17] suggested that metformin therapy increases the likelihood of ovulation in women with PCOS. Additionally, recent studies indicate that although incretin therapy is not yet routinely used in PCOS (especially in infertility treatment in PCOS), it may be considered in overweight or obese women with insulin resistance, especially when routine/standard methods fail [18].

This study aimed to evaluate the concentrations of GIP, GLP-1, and DPP-4 and to examine their associations with fasting and post-oral glucose tolerance test (OGTT) glucose levels, fasting insulin levels, and selected serum sex hormone levels in women with PCOS as potential factors involved in the development and progression of the disease.

Given the limited research on the potential impact of tobacco smoke exposure on incretin hormones and DPP-4 levels in the context of PCOS, we also incorporate this variable into our analysis. Previous studies have clearly demonstrated that smokers tend to have elevated fasting glucose and insulin levels, as well as higher levels after an oral glucose tolerance test, contributing to increased IR and a higher risk of T2D [19,20,21]. Therefore, it was also essential to consider tobacco smoke exposure as a relevant factor in the present study.

## 2. Results

### 2.1. Results of Studied Parameters in the Group of Women with or Without PCOS

The study groups included 60 women with PCOS and 28 women without PCOS, matched for age and body mass index (BMI). Firstly, we compared the investigated parameters between women with and without PCOS. The group of women with PCOS exhibited significantly higher concentrations of fasting glucose, fasting insulin, GIP, GLP-1, dehydroepiandrosterone sulphate (DHEA-S), total testosterone (tTest), free testosterone (fTest), and anti-Müllerian hormone (AMH), as well as higher Homeostatic Model Assessment for Insulin Resistance (HOMA-IR) and free androgen index (FAI) values, and an elevated LH/FSH ratio compared to women without PCOS. Women with PCOS were also characterized by lower glucose/insulin (G/I) ratio and the Quantitative Insulin Sensitivity Check Index (QUICKI) value. The characteristics of women with PCOS and women without PCOS are shown in Table 1.

### 2.2. Results of Studied Parameters in the Group of Smoking and Non-Smoking Women with PCOS

Table 2 presents the results obtained for smoking and non-smoking women with PCOS. There were no statistically significant differences in age and BMI between the two subgroups. However, fasting glucose levels were significantly higher in smoking women with PCOS compared to non-smoking women. The concentration of DPP-4 was approximately two times lower in the smoking group, while the levels of AMH, androstenedione (AD), and cortisol were reduced compared to the non-smoking group. No other significant differences were observed.

### 2.3. The Studied Parameters in Women with PCOS Stratified According to Body Weight or IR

The next step of the study was to analyze how exposure to tobacco smoke influences studied parameters in subgroups of women with normal or elevated weight (Table 3). Among these parameters, only DPP-4 concentrations showed statistically significant differences between non-smokers and smokers in both groups of PCOS women, those with normal and those with elevated weight. No other significant differences (*p* < 0.05) were observed between these subgroups.

We also compared the studied parameters between non-smoking women with normal and elevated weight as well as between smoking women with normal and elevated weight. In non-smoking women, approximately fourfold higher concentrations of GIP and GLP-1 were observed in those with elevated weight compared to those with normal weight. Additional significant differences were found in post-OGTT glucose concentrations, insulin levels, and indices of insulin resistance and sensitivity, including HOMA-IR, QUICKI, and G/I ratio. However, fasting glucose concentration remained comparable between the two groups. A similar pattern of statistically significant differences was observed within the smoking subgroup of women with PCOS. The results are summarized in Table 3.

The final stratification was based on HOMA-IR values (Table 4). We compared the studied parameters between non-smoking women with HOMA-IR < 2.0 and ≥2.0, as well as between smoking women with HOMA-IR < 2.0 and ≥2.0. A significant difference in DPP-4 concentrations was observed only between smokers and non-smokers within the PCOS subgroup with HOMA-IR <2.0.

Further analysis, also presented in Table 4, compares parameters between non-smokers with HOMA-IR < 2.0 and those with HOMA-IR ≥ 2.0, and similarly between smokers with HOMA-IR < 2.0 and ≥2.0. Among non-smoking women, significant differences were found in all listed parameters except DPP-4 concentration. In contrast, within the subgroup of smoking women, all parameters except glucose concentration differed significantly between those with HOMA-IR < 2.0 and those with HOMA-IR ≥ 2.0.

### 2.4. Diagnostic Value of GIP, GLP-1, and DPP-4

Receiver Operating Characteristic (ROC) analysis with area under curve (AUC) calculation was performed to evaluate the ability of GIP, GLP-1, and DDP-4 to discriminate women with PCOS from the control groups. Additionally, we graphically present the diagnostic value of these parameters in women with PCOS, comparing subgroups based on tobacco smoke exposure, body weight (normal and elevated weight), and IR status.

In the overall cohort (A), both GIP and GLP-1 concentrations demonstrated statistically significant AUCs (*p* < 0.01), indicating good discriminatory ability, whereas DPP-4 (AUC~0.50) was not useful as a classifier of PCOS. In the PCOS group stratified by smoking status (B), no significant discriminatory values were observed. Moreover, a strong negative z-score for DPP-4 suggests a potential reverse trend or poor classification performance.

Very high AUCs for GIP (0.939) and GLP-1 (0.942) were observed in the PCOS group stratified by body weight (C), indicating excellent predictive value, while DPP-4 remained still non-significant. Similarly, high AUC (0.89–0.90) for GIP and GLP-1 were found in the PCOS group stratified by HOMA-IR (D), further supporting their strong predictive utility. In contrast, the modest AUC for DPP-4 (0.648) with a marginally significant *p*-value (0.043) reinforces that DPP-4 is not a reliable classifier in the context of PCOS, body weight, or IR. The results are presented in Figure 1, Figure 2 and Figure 3 and Table 5.

The order of the ROC graphs corresponds to the order of the parameters presented in Table 5 (Figure 1, Figure 2 and Figure 3).

### 2.5. Correlations

We also examined potential correlation between studied parameters. In the entire group of women with PCOS, the highest correlation coefficient (0.99; *p* < 0.000) was noted between GIP and GLP-1. No significant correlations were found between the levels of incretin hormones or DPP-4 concentrations and age. In contrast, anthropometric parameters (BMI, WHR, WHtR) were positively associated with GIP and GLP-1 levels. All parameters related to glucose metabolism also showed significant correlations with incretin hormone levels. The weakest correlation (lowest r value) was observed between GIP or GLP-1 levels and post-OGTT glucose (glucose at 120 min), compared to fasting glucose (glucose at 0 min). Cotinine concentration was significantly correlated only with DPP-4 concentration, and this relationship was negative. All correlations are summarized in Table 6.

## 3. Discussion

Disorders in glucose metabolism are among the most common and clinically significant comorbidities associated with PCOS. Insulin resistance affects approximately 35–80% of women with PCOS, often independently of BMI [22]. A meta-analysis found that insulin sensitivity is, on average, 27% lower in women with PCOS compared to healthy controls [23]. The regulation of glucose homeostasis involves several components, including the incretin hormones (GIP, GLP-1) and DPP-4. Therapeutic agents targeting this pathway, such as metformin, GLP-1 receptor agonists, GIP analogues, and DPP-4 inhibitors, are widely used to improve glycemic control [24].

Given this background, our study aimed to assess serum concentrations of GIP, GLP-1, and DPP-4 in women with PCOS and to explore their associations with metabolic and hormonal parameters, particularly in the context of tobacco smoke exposure, body weight, and IR.

We first compared these parameters between women with and without PCOS. Notably, women with PCOS showed significantly higher fasting glucose and insulin levels, as well as elevated concentrations of GIP and GLP-1. A particularly strong positive correlation was observed between GIP and GLP-1 levels (0.99; *p* < 0.0001). However, existing literature on incretin levels in PCOS is inconsistent. Some studies report reduced fasting and postprandial GLP-1 levels in women with PCOS [25], while others found no significant difference [26]. Milewicz et al. [27] suggested that hyperandrogenism, especially in lean PCOS patients, may influence incretin secretion. Our findings support this, as we observed positive associations between GIP and GLP-1 levels and both free testosterone and FAI, as well as negative correlations with SHBG. This pattern suggests a potential link between elevated incretin levels and hyperandrogenism.

GIP and GLP-1 have also been proposed as biomarkers for metabolic syndrome and hormonal dysfunction. Seon et al. [28] identified GLP-1 as a potential biomarker for metabolic syndrome, citing its role in enhancing insulin secretion, improving insulin sensitivity, promoting satiety, slowing gastric emptying, and aiding in weight loss [11]. Chang et al. [29] suggested that GIP may serve as a novel biomarker for PCOS and play a role in its pathogenesis. Moffett and Naughton [30] further proposed that incretin hormones may influence female fertility and could offer new therapeutic avenues in reproductive disorders such as PCOS.

Our results support these hypotheses. We observed strong correlations between both incretins and measures of IR (HOMA-IR, QUICKI), adiposity indices (BMI, WHR, WHtR), and fasting glucose and insulin levels. The associations were notably stronger for fasting glucose than for post-OGTT glucose, suggesting a more prominent role of incretins in early-phase glucose regulation. Previous studies have also reported a blunted GIP response to OGTT in obese women with PCOS, possibly reflecting a compensatory upregulation of insulinotropic signaling [31].

In examining the influence of overweight and obesity, we found significantly elevated levels of GIP and GLP-1 in women with PCOS and elevated weight, along with higher insulin and IR indices. Similar differences were observed when stratifying PCOS patients by HOMA-IR (<2.0 and ≥2.0), supporting the hypothesis that early metabolic disturbances precede overt hyperglycemia. Literature suggests that obesity may be associated with GIP hypersecretion as a compensatory mechanism in IR [32], potentially indicating incretin resistance. Taken together, our findings indicate that higher concentrations of incretin hormones were observed in PCOS women who are overweight or obese, which also highlights that body weight loss can significantly improve glucose regulation parameters such as GIP and GLP-1.

In line with these observations, Vrbikova et al. [33] reported elevated GIP and unchanged early-phase GLP-1 levels in lean PCOS women compared to controls, with significantly lower GLP-1 levels at 180 min post-OGTT. Differences in GLP-1 levels across studies may be partially explained by differences in body weight. In our cohort, both GIP and GLP-1 levels were highest in overweight or obese women with PCOS and showed strong positive correlations with BMI, WHR, and WHtR.

Importantly, our data suggest that IR and body weight exert a greater influence on incretin levels than tobacco smoke exposure. While BMI and IR strongly correlate with incretin concentrations [34], we found no significant differences in GIP and GLP-1 levels between smokers and non-smokers. Cotinine levels were also uncorrelated with incretin concentration, suggesting a minimal role of tobacco exposure in regulating these hormones. This finding aligns with previous studies reporting inconsistent or negligible effects of smoking on incretin levels [35,36,37]. In a pilot study involving 22 individuals, no significant changes in GIP and GLP-1 concentrations were observed three months after smoking cessation [35]. Also Grøndahl et al. [19], even if they observed significant changes in fasting glucagon levels in smokers, no significant differences in postprandial GLP-1 and GIP levels were observed between smokers and nonsmokers. Similarly, Stadler et al. [36] found no significant differences in fasting GLP-1 levels between current smokers and individuals who had quit smoking for more than three months. However, Driva et al. [37] reported that while 2.5 months of smoking cessation did not affect GLP-1 levels in individuals with T2D, a duration of four months was sufficient to observe a significant increase. Given these discrepancies, further investigation in larger cohorts is warranted.

However, tobacco exposure impacted certain hormonal parameters. Smoking PCOS women had significantly lower AMH, AD, and morning cortisol levels. Although some studies, like that of Waylen et al. [38], found no such associations, our findings underscore the potential endocrine-disrupting effects of tobacco smoke.

Notably, DPP-4 concentrations appeared to be particularly sensitive to tobacco exposure. Smoking PCOS women had approximately two-fold lower DPP-4 levels than non-smokers. Moreover, DPP-4 levels were nearly five times lower in smoking women with HOMA-IR<2.0 compared to those with HOMA-IR ≥ 2.0. A significant negative correlation was also observed between DPP-4 and cotinine levels (r= -0.41; *p* < 0.001). Interestingly, DPP-4 did not significantly correlate with GIP or GLP-1 (r = 0.12; *p* < 0.369 and r = 0.11; *p* < 0.286, respectively), suggesting that DPP-4 may be regulated independently of incretin hormones and more directly influenced by external factors such as smoking.

We also found that DPP-4 concentration did not differ between PCOS and non-PCOS women, nor between normal and elevated-weight PCOS subgroups. In contrast to GIP and GLP-1, these results indicate that DPP-4 may be more responsive to environmental factors such as smoking rather than intrinsic metabolic abnormalities associated with PCOS.

Age, as a known risk factor for glucose intolerance, IR, and T2D [39,40], showed no significant correlation with GIP, GLP-1, or DPP-4 levels in our study. This may be due to the relatively narrow age range (20–30 years) of our study population, within which age-related metabolic changes are likely minimal.

ROC analysis suggested the excellent diagnostic potential for both GIP and GLP-1, with AUC values around 0.89–0.90 in PCOS women stratified by body weight and IR. These findings support the potential utility of incretin-based therapies in managing metabolic dysfunction in PCOS. Preclinical studies have shown that GIP receptor antagonism can reverse obesity and IR in animal models [41]. Although the role of GIP in reproduction is not well-established, some evidence suggests it may impair FSH-induced progesterone synthesis or reduce FSH levels via central mechanisms [42,43]. For instance, Gosman et al. [42] found that intraventricular injection of GIP in rats led to a reduction in serum FSH levels, suggesting a possible negative effect on fertility.

GLP-1 RAs have shown superior efficacy compared to metformin in promoting weight loss, improving menstrual frequency, and enhancing spontaneous pregnancy rates in women with PCOS [44]. In a study by Tzotzas et al. [45], GLP-1 RAs, alone or in combination with metformin, significantly reduced body weight reduction and testosterone levels and improved insulin sensitivity. Another study report improved menstrual regularity with GLP-1 RAs compared to metformin or placebo [13]. Immunohistochemical studies have demonstrated GLP-1-positive axons projecting to GnRH neurons in the hypothalamus, suggesting a role in central reproductive hormone regulation [46].

Animal models have further shown that liraglutide, a GLP-1 RA, either alone or in combination with metformin, improves metabolic, hormonal, and reproductive outcomes in PCOS, including cycle regularity, ovarian morphology, and oxidative stress [47]. DPP-4 inhibitors such as sitagliptin have also demonstrated potential for improving beta-cell function and may serve as adjunct therapies [48].

A recent network meta-analysis involving 1476 participants concluded that combining standard therapy with GLP-1 RAs offers superior metabolic and hormonal outcomes in PCOS compared to standard therapy alone [49]. Moreover, dual GLP-1/GIP RAs may offer greater benefits than GLP-1 monotherapy [50]. In our study, the similar correlation patterns of GIP or GLP-1 with metabolic and hormonal markers suggest coordinated roles in PCOS pathophysiology and reinforce the potential of dual-agonist therapy.

While GIP and GLP-1 concentrations appear to be useful markers of glucose regulation in women with PCOS, exposure to tobacco smoke and elevated cotinine levels may act as confounding factors. Depending on the degree of exposure, individualized therapeutic strategies involving GLP-1 and/or GIP analogues or DPP-4 inhibitors may be warranted.

## 4. Materials and Methods

### 4.1. Materials

The study was conducted among a cohort of 88 women admitted to the Gynecological Endocrinology Clinic at the Silesian Medical University in Katowice, Poland. The diagnosis of PCOS was based on the Rotterdam criteria (2003), requiring the presence of at least two of the following three features: ovulatory dysfunction, biochemical and/or clinical hyperandrogenism, and polycystic ovarian morphology [51]. Sixty women met the diagnostic criteria for PCOS. Among them, 37 were non-smokers and 23 were smokers. The control group comprised 28 non-smoking women who were initially referred for suspected PCOS; however, hormonal and imaging evaluations excluded the diagnosis, and their hormonal profiles were within normal limits. Participants with hyperprolactinemia, type 1 or type 2 diabetes, hypertension, Cushing’s syndrome, adrenal tumors, or alcohol abuse were excluded from both the study and control groups. More details describing the PCOS cohort were published previously [52]. Ethical approval was obtained from the Bioethical Committee of Wroclaw Medical University (approval number 222/2024).

### 4.2. Methods

Anthropometric measurements (height, weight, waist circumference, and hip circumference) and the modified Ferriman-Gallwey (mFG) scale of hirsutism were performed during hospitalization. Hormonal profiles (including LH, follicle-stimulating hormone (FSH), DHEA-S, tTest; fTest, AMH, and AD) and the concentrations of fasting and post-OGTT glucose and insulin levels were also assessed in the hospital laboratory using routine procedures described earlier in our paper [52]. The data about exposure to tobacco smoke was received from individual interviews and confirmed by determination of cotinine level in serum using a commercially available test (Cotinine ELISA, Ref. No. CO096D, Calbiotech Inc., El Cajon, CA, USA), which was also described in our previous paper [52]. According to the findings reported by Duque et al. [53], self-reported smoking status and serum cotinine levels ≥ 10 ng/mL are accurate, complementary, and reliable indicators for determining an individual’s smoking status. These methods are particularly suitable for use in large-scale population studies and multicenter research involving serum samples. Based on these criteria, we stratified the women in our study into smoking and non-smoking groups.

The commercially available kits from Cusabio (Houston, TX, USA) were used to measure the concentration of GIP (Human glucose-dependent insulin releasing polypeptide (GIP) ELISA Kit Cat. Number: CSB-E08484h), GLP-1 (Human glucagon-like peptide-1 (GLP-1), ELISA Kit Cat. Number: CSB-E08119h), and DPP-4 (Human Dipeptidyl Peptidase IV (DPPⅣ) ELISA Kit Cat. Number: CSB-E08518h).

IR was assessed using the Homeostatic Model Assessment for Insulin Resistance, calculated with the standard formula:HOMA−IR=Fasting glucosemmolL×fasting insulinµIUmL22.5

In accordance with the European Group for the Study of Insulin Resistance, which recommends a threshold of >2.0 for diagnosing IR in young Caucasian women with polycystic ovary syndrome, we adopted this value to define IR in our study [54].

To assess insulin sensitivity, the Quantitative Insulin Sensitivity Check Index was calculated using the following formula:QUICKI                                                 =1logfasting glucosemg100mL+logfasting insulinµIUmL

A QUICKI value ≤0.33 was considered indicative of IR [55].

Body mass index (BMI), waist-to-hip ratio (WHR), waist-to-height ratio (WHtR), free androgen index (FAI), glucose-to-insulin ratio (G/I), and luteinizing hormone to follicle-stimulating hormone (LH/FSH) ratio were also determined using standard formulas. The cut-off values were defined as follows: 25.0 for BMI [56], 0.8 for WHR [57], and 0.5 for WHtR [58].

### 4.3. Statistical Analysis

All analyses were performed using the Polish version 13.3 of the Statistica Software Package. Data are presented as mean value and standard deviation (X ± SD) and median with the first (Q1) and third (Q3) quartiles. The Shapiro-Wilk test was used to assess the normality of variable distributions, and Levene’s test was applied to evaluate the homogeneity of variances. When assumptions of normality and equal variance were not met, differences between two groups were analyzed using the non-parametric Mann–Whitney U test. Correlations were examined using Spearman’s rank-order correlation coefficient. Statistical significance was set at *p* < 0.05.

## 5. Conclusions

In summary, our findings indicate that GIP and GLP-1 concentrations are closely associated with metabolic and hormonal disturbances in women with PCOS. These associations are strongly influenced by IR and adiposity, while tobacco exposure appears to have a more substantial effect on DPP-4 levels. Given these results, incretin hormones hold promise both as biomarkers and therapeutic targets in PCOS. However, these findings are preliminary and require validation in a larger and more diverse population.

## Figures and Tables

**Figure 1 ijms-26-07097-f001:**
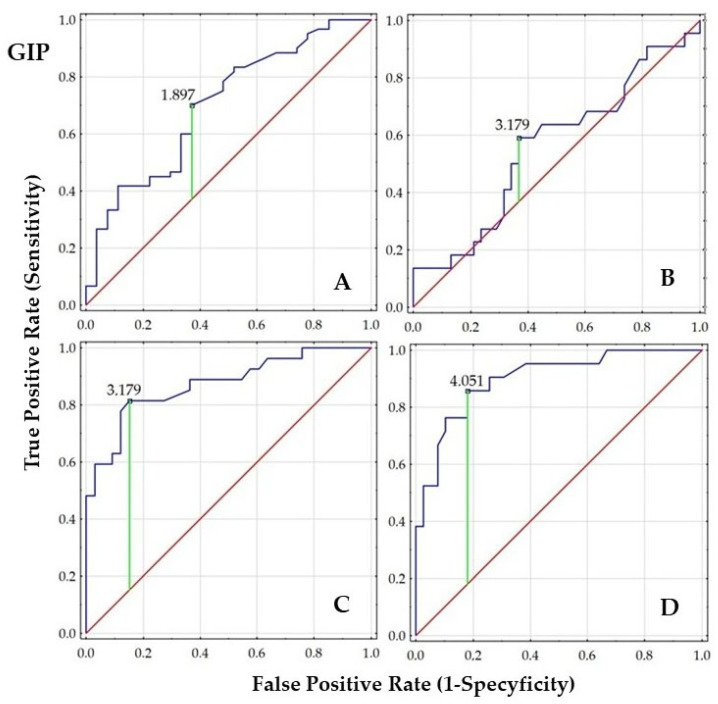
Comparison of ROC analysis for GIP concentration between the subgroups ((**A**)—women with or without PCOS; (**B**)—smoking women with PCOS vs. non-smoking women with PCOS; (**C**)—PCOS women with BMI < 25.0 vs. PCOS women with BMI ≥ 25.0; (**D**)—PCOS women with HOMA-IR < 2.0 vs. PCOS women with HOMA-IR ≥ 2.0); (navy line—ROC curve representing the performance of the classifier at various threshold settings; green line—optimal classification threshold based on Youden’s Index indicating the best trade-off between sensitivity and specificity; red line—line of no discrimination, representing random classification).

**Figure 2 ijms-26-07097-f002:**
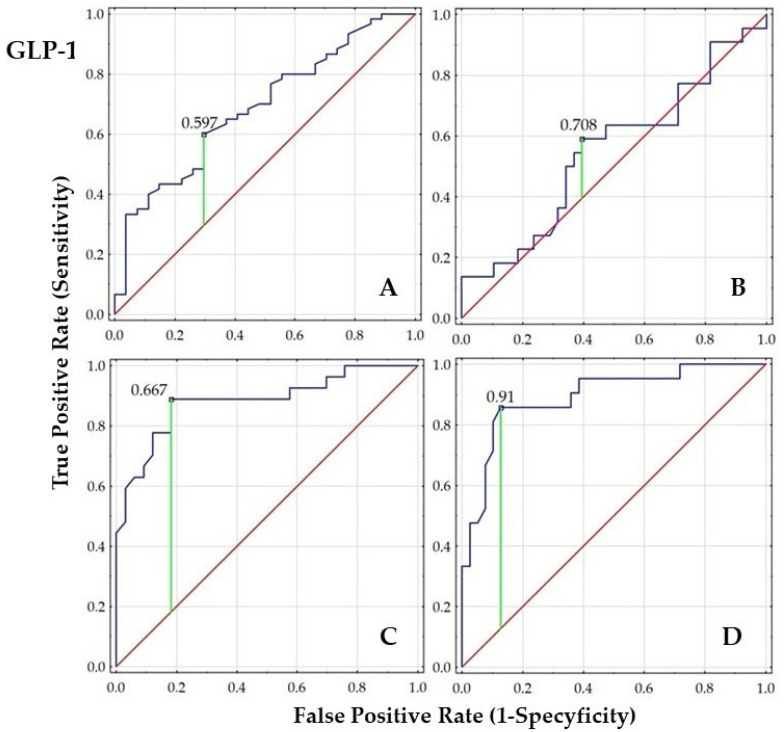
Comparison of ROC analysis for GLP-1 concentration in the studied groups and subgroups ((**A**)—women with or without PCOS; (**B**)—smoking women with PCOS vs. non-smoking women with PCOS; (**C**)—PCOS women with BMI < 25.0 vs. PCOS women with BMI ≥ 25.0; (**D**)—PCOS women with HOMA-IR < 2.0 vs. PCOS women with HOMA-IR ≥ 2.0); (navy line—ROC curve representing the performance of the classifier at various threshold settings; green line—optimal classification threshold based on Youden’s Index indicating the best trade-off between sensitivity and specificity; red line—line of no discrimination, representing random classification).

**Figure 3 ijms-26-07097-f003:**
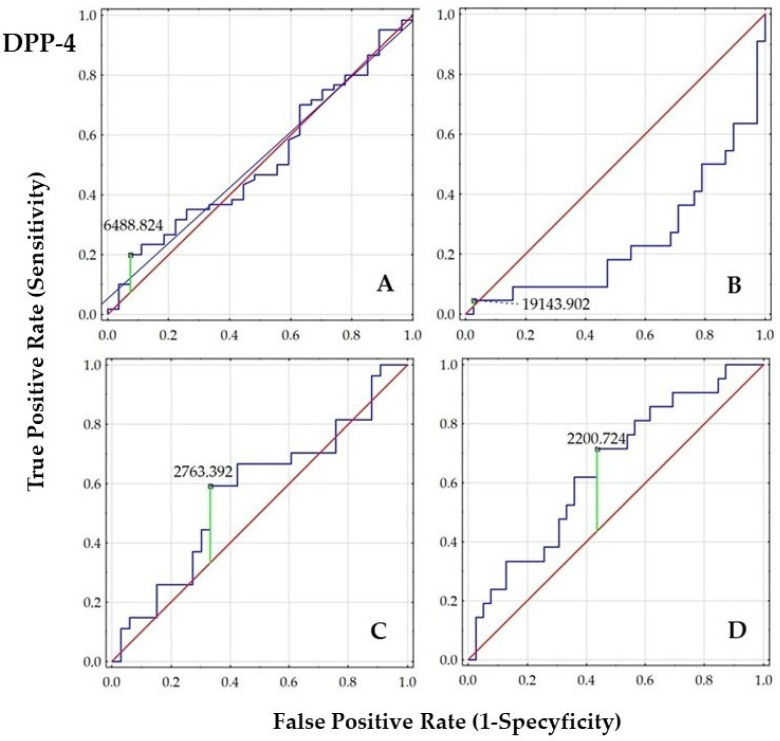
Comparison of ROC analysis for DPP-4 concentration between the subgroups ((**A**)—women with or without PCOS; (**B**)—smoking women with PCOS vs. non-smoking women with PCOS; (**C**)—PCOS women with BMI < 25.0 vs. PCOS women with BMI ≥ 25.0; (**D**)—PCOS women with HOMA-IR < 2.0 vs. PCOS women with HOMA-IR ≥ 2.0); (navy line—ROC curve representing the performance of the classifier at various threshold settings; green line—optimal classification threshold based on Youden’s Index indicating the best trade-off between sensitivity and specificity; red line—line of no discrimination, representing random classification).

**Table 1 ijms-26-07097-t001:** Characteristics of women with PCOS and women without PCOS, with particular emphasis on glucose regulation.

Parameters	PCOS Women	Women without PCOS	*p*-Value
	n = 60	n = 28	
Age (years)	25.50 ± 3.83 24.50 (22.00–28.00)	26.71 ± 5.87 27.00 (22.00–31.00)	0.480
BMI (kg/m^2^)	26.13 ± 6.74 23.93 (20.76–30.86)	22.62 ± 4.58 21.63 (20.57–23.03)	0.21
mFG score of hirsutism	5.87 ± 4.63 6.00 (1.50–9.00)	NA	NA
Glucose 0′ (mg/dL)	85.31 ± 6.75 84.35 (80.80–89.30)	81.65 ± 5.19 80.50 (78.50–86.80)	0.049
Glucose 120′ (mg/dL)	118.41 ± 41.95 111.77 (89.20–130.00)	94.40 ± 27.06 93.50 (80.20–103.00)	0.066
Insulin 0′ (mU/mL)	9.50 ± 6.86 7.14 (5.18–12.15)	5.34 ± 2.52 5.93 (3.49–6.77)	0.002
G/I ratio	12.55 ± 6.83 11.78 (7.42–16.09)	20.95 ± 13.52 14.15 (11.86–26.46)	0.003
HOMA-IR	2.06 ± 1.70 1.51 (1.04–2.68)	1.09 ± 0.52 1.18 (0.70–1.41)	0.002
QUICKI	0.36 ± 0.03 0.36 (0.33–0.38)	0.39 ± 0.04 0.37 (0.36–0.42)	0.002
GIP (ng/mL)	6.75 ± 17.75 2.46 (1.67–6.67)	2.70 ± 4.18 1.54 (1.03–2.80)	0.003
GLP-1 (ng/mL)	1.49 ± 2.44 0.66 (0.41–1.41)	0.65 ± 0.94 0.43 (0.27–0.69)	0.005
DPP-4 (µg/mL)	4.31 ± 5.53 2.37 (1.15–4.56)	3.86 ± 5.10 2.64 (1.23–4.22)	0.819
LH/FSH ratio	1.44 ± 0.70 1.26 (1.00–1.65)	0.98 ± 0.83 0.79 (0.54–1.18)	0.001
Cortisol (µg/dL)	12.84 ± 3.91 12.65 (9.86–16.00)	14.31 ± 5.66 13.75 (10.80–17.50)	0.381
DHEA-S (µg/mL)	320.63 ± 135.59 294.00 (222.00–386.50)	252.51 ± 143.42 230.00 (164.00–295.00)	0.015
SHBG (nmol/L)	56.54 ± 35.99 39.95 (28.40–55.50)	68.91 ± 37.07 62.50 (50.20–71.60)	0.118
tTest (ng/mL)	0.41 ± 0.16 0.36 (0.30–0.53)	0.30 ± 0.22 0.26 (0.16–0.32)	0.004
fTest (pg/mL)	3.20 ± 2.78 2.11 (1.46–4.00)	2.22 ± 2.25 1.45 (0.71–2.06)	0.042
AD (ng/mL)	2.48 ± 0.90 2.33 (1.89–2.83)	2.06 ± 0.73 1.86 (1.59–2.20)	0.085
FAI	3.74 ± 3.15 2.78 (1.60–4.40)	1.83 ± 1.54 1.2 (0.72–2.61	0.005
AMH (ng/mL)	6.28 ± 3.06 5.98 (3.93–7.78)	3.18 ± 2.18 2.47 (1.63–4.49)	0.001

**Table 2 ijms-26-07097-t002:** Characteristics of non-smoking and smoking women with PCOS and women without PCOS, with particular emphasis on glucose regulation.

Parameters	Smoking with PCOS	Non-Smoking with PCOS	*p*-Value
	n = 23	n = 37	
Age (years)	26.46 ± 4.47 25.50 (23.00–29.00)	24.51 ± 4.79 24.50 (22.00–27.00)	0.262
BMI (kg/m^2^)	27.30 ± 7.57 25.55 (21.01–31.25)	25.15 ± 7.14 23.66 (20.42–29.49)	0.381
mFG score of hirsutism	5.48 ± 3.93 6.00 (2.00–8.00)	6.02 ± 5.23 5.00 (1.50–9.00)	0.837
Glucose 0′ (mg/dL)	87.96 ± 7.96 86.65 (81.00–92.00)	83.45 ± 5.20 83.50 (79.80–87.50)	0.040
Glucose 120′ (mg/dL)	118.41 ± 41.95 111.00 (95.50–133.00)	106.67 ± 33.41 96.65 (85.40–125.00)	0.289
Insulin 0′ (mU/mL)	9.55 ± 7.76 7.52 (5.37–11.40)	9.72 ± 7.00 6.97 (4.98–12.50)	0.958
G/I ratio	13.61 ± 8.61 11.44 (7.52–16.65)	11.93 ± 5.59 11.91 (7.40–16.03)	0.790
HOMA-IR	2.17 ± 2.11 1.54 (1.09–2.41)	2.07 ± 1.60 1.50 (0.98–2.76)	0.885
QUICKI	0.36 ± 0.04 0.36 (0.34–0.38)	0.36 ± 0.03 0.36 (0.33–0.39)	0.897
GIP (ng/mL)	9.75 ± 17.77 3.64 (1.69–7.03)	5.00 ± 5.69 2.41 (1.64–6.31)	0.460
GLP-1 (ng/mL)	2.11 ± 3.69 0.83 (0.42–1.42)	1.13 ± 1.20 0.63 (0.40–1.41)	0.527
DPP-4 (µg/mL)	2.52 ± 4.12 1.37 (0.62–2.20)	5.34 ± 6.02 3.06 (1.81–6.49)	0.001
LH/FSH ratio	1.51 ± 0.84 1.19 (0.96–1.79)	1.40 ± 0.60 1.28 (1.00–1.58)	0.958
Cortisol (µg/dL)	11.43 ± 3.80 11.00 (8.99–13.70)	13.65 ± 3.78 14.40 (11.00–17.00)	0.036
DHEA-S (µg/mL)	306.14 ± 161.93 252.00 (192.00–362.00)	326.50 ± 127.11 301.00 (223.00–397.00)	0.176
SHBG (nmol/L)	49.56 ± 34.26 39.95 (28.40–55.50)	62.85 ± 43.64 58.55 (34.40–74.70)	0.103
tTest (ng/mL)	0.39 ± 0.17 0.35 (0.29–0.57)	0.41 ± 0.17 0.38 (0.31–0.51)	0.549
fTest (pg/mL)	3.26 ± 3.18 1.98 (1.47–4.11)	3.14 ± 2.58 2.18 (1.51–3.88)	0.825
AD (ng/mL)	2.11 ± 0.60 2.13 (1.65–2.57)	2.66 ± 1.04 2.48 (1.90–3.39)	0.048
FAI	4.28 ± 3.57 3.48 (1.54–5.67)	3.55 ± 3.12 2.73 (1.74–3.95)	0.414
AMH (ng/mL)	5.12 ± 2.64 5.20 (3.19–6.39)	6.95 ± 3.53 6.66 (3.98–9.34)	0.029

**Table 3 ijms-26-07097-t003:** The influence of exposure to tobacco smoke in the subgroups of women with PCOS with normal (BMI < 25.0, WHR < 0.8, and WHtR < 0.5) or elevated body weight (BMI ≥ 25.0, WHR ≥ 0.8, and WHtR ≥ 0.5).

Variables	Normal Weight Women with PCOS		Elevated Weight Women with PCOS	
	Non-Smoking	Smoking	Non-Smoking	Smoking
	n = 23	n = 10	n = 15	n = 12
Glucose 0′ (mg/dL)	83.43 ± 5.57 83.45 (79.65–86.75)	85.14 ± 5.09 86.250 (80.90–87.60)	85.34 ± 5.83 86.10 (80.85–90.20)	90.31 ± 9.29 88.50 (82.25–96.85)
Glucose 120′ (mg/dL)	100.61 ± 22.87 95.40 (84.75–113.50)	110.06 ± 21.53 111.00 (95.50–124.00)	122.83 ± 38.47 118.00 (90.85–148.50) ^2^	125.36 ± 53.52 111.00 (91.65–137.00)
Insulin 0′ (mU/mL)	6.44 ± 2.56 6.11 (4.33–7.29)	5.66 ± 2.92 5.02 (3.52–7.38)	14.40 ± 7.14 12.85 (9.66–18.50) ^1^	12.71 ± 9.10 10.65 (7.11–14.75) ^3^
G/I ratio	14.86 ± 4.51 13.74 (11.75–18.03)	19.06 ± 9.75 17.63 (11.74–26.99)	7.45 ± 3.88 6.53 (4.48–10.24) ^1^	9.07 ± 3.78 8.89 (6.61–12.35) ^3^
HOMA-IR	1.35 ± 0.62 1.26 (0.85–1.56)	1.18 ± 0.58 1.09 (0.81–1.50)	3.07 ± 1.63 2.76 (2.00–3.85) ^1^	2.99 ± 2.56 2.35 (1.55–3.26) ^3^
QUICKI	0.37 ± 0.02 0.37 (0.36–0.40)	0.38 ± 0.03 0.38 (0.36–0.40)	0.33 ± 0.03 0.33 (0.31–0.35) ^1^	0.34 ± 0.03 0.34 (0.32–0.36) ^3^
GIP (ng/mL)	2.20 ± 1.46 2.05 (1.23–2.41)	2.40 ± 1.53 1.92 (1.33–3.18)	9.30 ± 7.04 7.44 (4.05–13.85) ^1^	15.89 ± 22.59 5.95 (3.64–22.08) ^3^
GLP-1 (ng/mL)	0.54 ± 0.33 0.50 (0.30–0.62)	0.55 ± 0.33 0.45 (0.33–0.78)	2.04 ± 1.47 1.69 (0.82–3.00) ^1^	3.41 ± 4.67 1.31 (0.80–4.85) ^3^
DPP-4 (µg/mL)	4.87 ± 6.66 2.42 (1.34–5.69)	1.88 ± 2.40 1.55 (0.51–1.93) *	6.08 ± 5.00 4.13 (2.76–7.68)	3.05 ± 5.20 1.13 (0.65–3.14) **

* *p* < 0.007 when compared to non-smoking women with PCOS; ** *p* < 0.043 when compared to smoking women with PCOS. ^1^
*p* < 0.001 when compared to non-smoking women with PCOS and normal weight; ^2^
*p* < 0.033 when compared to non-smoking women with PCOS and normal weight; ^3^
*p* < 0.003 when compared to smoking women with PCOS and normal weight.

**Table 4 ijms-26-07097-t004:** Influence of tobacco smoke exposure in women with PCOS, stratified by insulin sensitivity (HOMA-IR < 2.0) or insulin resistance (HOMA-IR ≥ 2.0).

Variables	HOMA-IR < 2.0 in Women with PCOS		HOMA-IR ≥2.0 in Women with PCOS	
	Non-Smoking	Smoking	Non-Smoking	Smoking
	n = 25	n = 14	n = 13	n = 8
Glucose 0′ (mg/dL)	81.78 ± 4.80 82.30 (77.60–84.40)	85.19 ± 4.64 86.25 (81.00–87.60)	87.59 ± 4.80 88.90 (83.70–91.00) ^2^	92.80 ± 10.38 92.85 (83.30–100.00)
Glucose 120′ (mg/dL)	99.85 ± 24.57 92.90 (83.50–115.00)	105.90 ± 23.12 107.50 (77.60–124.00)	123.47 ± 34.17 108.00 (95.90–148.00) ^2^	140.23 ± 58.54 125.00 (105.50–154.50)
Insulin 0′ (mU/mL)	5.90 ± 1.76 6.08 (4.38–6.89)	5.69 ± 2.07 5.79 (4.10–7.38)	16.36 ± 6.44 13.40 (12.50–18.70) ^1^	16.31 ± 9.53 12.75 (11.30–16.60) ^3^
G/I ratio	14.98 ± 4.19 13.74 (12.01–17.92)	17.60 ± 8.38 14.51 (11.74–19.21)	6.07 ± 2.17 6.05 (4.48–7.40) ^1^	6.63 ± 2.10 6.85 (5.46–8.23) ^3^
HOMA-IR	1.20 ± 0.38 1. 17 (0.86–1.48)	1.19 ± 0.43 1.22 (0.83–1.50)	3.53 ± 1.46 2.91 (2.76–3.87) ^1^	3.87 ± 2.79 2.95 (2.41–3.68) ^3^
QUICKI	0.38 ± 0.02 0.37 (0.36–0.94)	0.38 ± 0.03 0.37 (0.36–0.40)	0.32 ± 0.02 0.33 (0.31–0.33) ^1^	0.32 ± 0.02 0.33 (0.32–0.34) ^3^
GIP (ng/mL)	2.24 ± 1.47 2.05 (1.28–2.41)	3.34 ± 3.00 2.31 (1.59–4.15)	10.32 ± 7.00 8.87 (5.64–13.85) ^1^	20.97 ± 26.53 6.23 (4.49–33.28) ^4^
GLP-1 (ng/mL)	0.55 ± 0.33 0.50 (0.32–0.66)	0.78 ± 0.70 0.55 (0.34–0.89)	2.25 ± 1.45 1.81 (1.25–3.00) ^1^	4.45 ± 5.49 1.31 (1.05–7.22) ^4^
DPP-4 (µg/mL)	5.10 ± 6.34 3.19 (1.89–5.69)	1.04 ± 0.65 * 1.05 (0.51–1.58)	5.81 ± 5.55 2.93 (1.81–7.15)	5.11 ± 6.17 3.14 (1.44–6.00) ^4^

* *p* < 0.001 when compared to non-smoking women with PCOS; ^1^
*p* < 0.001 when compared to non-smoking PCOS women with HOMA-IR < 2.0; ^2^
*p* < 0.030 when compared to non-smoking PCOS women with HOMA-IR < 2.0; ^3^
*p* < 0.001 when compared to smoking PCOS women with HOMA-IR < 2.0; ^4^
*p* < 0.013 when compared to smoking PCOS women with HOMA-IR < 2.0.

**Table 5 ijms-26-07097-t005:** AUC values with corresponding statistics for GIP, GLP-1, and DPP-4 concentrations across examined groups and various subgroups.

	**AUC**	**SE**	**AUC Lower 95%**	**AUC Upper 95%**	**z = (v1–0.5)/v2**	*p* **-Value**
Entire Cohort: PCOS vs. Controls (A)						
GIP (ng/mL)	0.699	0.061	0.58	0.818	3.278	0.001
GLP-1 (ng/mL)	0.688	0.06	0.571	0.804	3.152	0.002
DPP-4 (µg/mL)	0.516	0.066	0.387	0.646	0.243	0.808
PCOS Group Stratified by Smoking Status (B)						
GIP (ng/mL)	0.559	0.079	0.404	0.713	0.744	0.457
GLP-1 (ng/mL)	0.55	0.079	0.394	0.705	0.625	0.532
DPP-4 (µg/mL)	0.244	0.067	0.112	0.376	−3.798	0.001
PCOS Group Stratified by Body Weight (C)						
GIP (ng/mL)	0.939	0.035	0.871	1	12.705	0.001
GLP-1 (ng/mL)	0.942	0.036	0.871	1	12.194	0.001
DPP-4 (µg/mL)	0.579	0.078	0.426	0.731	1.011	0.312
PCOS Group Stratified by Insulin Resistance (HOMA-IR <2.0 or ≥2.0) (D)						
GIP (ng/mL)	0.899	0.042	0.818	0.981	9.573	0.001
GLP-1 (ng/mL)	0.893	0.045	0.805	0.981	8.729	0.001
DPP-4 (µg/mL)	0.648	0.073	0.505	0.792	2.023	0.043

**Table 6 ijms-26-07097-t006:** Correlation coefficient between parameters in the entire group of women with PCOS.

	GIP (ng/mL)	GLP-1 (ng/mL)	DPP-4 (µg/mL)
GIP (ng/mL)	-	0.99; *p* < 0.000	NS
GLP-1 (ng/mL)	0.99; *p* < 0.001	-	NS
DPP-4 (µg/mL)	NS	NS	-
Age (years)	NS	NS	NS
Cotinine (ng/mL)	NS	NS	−0.41; *p* < 0.001
BMI (kg/m^2^)	0.64; *p* < 0.001	0.65; *p* < 0.001	NS
WHR	0.55; *p* < 0.001	0.56; *p* < 0.001	0.35; *p* < 0.007
WHtR	0.66; *p* < 0.001	0.67; *p* < 0.001	NS
Glucose 0′ (mg/dL)	0.35; *p* < 0.006	0.36; *p* < 0.005	NS
Glucose 120′ (mg/dL)	0.29; *p* < 0.022	0.31; *p* < 0.015	NS
Insulin 0′ (mU/mL)	0.82; *p* < 0.001	0.79; *p* < 0.001	NS
G/I	−0.82; *p* < 0.001	−0.79; *p* < 0.001	NS
HOMA-IR	0.80; *p* < 0.001	0.79; *p* < 0.001	NS
QUICKI	−0.80; *p* < 0.001	−0.79; *p* < 0.001	NS
LH/FSH	NS	NS	NS
Cortisol (µg/dL)	NS	NS	NS
DHEA-S (µg/mL)	NS	NS	NS
SHBG (nmol/L)	−0.72; *p* < 0.001	−0.71; *p* < 0.001	NS
tTest (ng/mL)	NS	NS	NS
fTest (pg/mL)	0.41; 0.001	0.42; *p* < 0.001	NS
AD (ng/mL)	NS	NS	NS
FAI	0.65; *p* < 0.001	0.65; *p* < 0.001	NS
AMH (ng/mL)	NS	NS	NS

## Data Availability

The data presented in this study are available upon request from the corresponding author.

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
