# Peer review of "Associations of Serum GIP, GLP-1, and DPP-4 with Metabolic and Hormonal Profiles and Tobacco Exposure in Women with Polycystic Ovary Syndrome"

_ijms, 2025, doi:10.3390/ijms26157097_

Round 1

Reviewer 1 Report

Comments and Suggestions for Authors

Bizon and colleagues investigate the serum concentrations of incretin hormones (GIP, GLP-1) and their regulatory enzyme (DPP-4) in women with polycystic ovary syndrome (PCOS), examining their associations with metabolic parameters, sex hormone levels, and exposure to tobacco smoke. Through detailed subgroup analyses stratified by smoking status, insulin resistance (HOMA-IR), and body weight, the study reveals elevated levels of GIP and GLP-1 in PCOS patients, along with reduced DPP-4 levels in smokers. It suggests that incretins may play significant roles in PCOS pathophysiology and may serve as diagnostic or therapeutic targets. Despite the significance of the content, the novelty could be better emphasized by more clearly distinguishing this study’s contribution relative to prior work. Another issues are shown below:

A) The design is generally sound, but the manuscript would benefit from clearer articulation of inclusion/exclusion criteria, recruitment procedures, and the rationale behind chosen cut-off points (e.g., HOMA-IR ≥2.0 for IR).

B) The cross-sectional nature of the study limits causal inference, yet some interpretations, particularly regarding smoking’s role in reducing DPP-4, lean toward causality. This should be tempered and discussed more critically in the discussion section.

C) The control group appears younger and leaner than the PCOS group, which may confound the interpretation of metabolic and hormonal differences. A more detailed comparison or statistical control for these variables is advised.

D) The manuscript would benefit from more mechanistic discussion regarding how smoking could alter DPP-4 expression or secretion, especially given the weak literature support and the lack of strong correlation between DPP-4 and incretin levels.

E) Figures 1–3 presenting ROC curves are informative, but resolution and figure legends need improvement. Table captions should also be made more descriptive for standalone comprehension.

F) Some content in the discussion repeats results verbatim or revisits points multiple times (e.g., DPP-4 and smoking), which reduces narrative flow.

G) All abbreviations should be defined at first mention in both the abstract and the main text. For example, “IR” appears in the abstract without definition.

Comments on the Quality of English Language

The manuscript is mostly well-written, though there are occasional grammatical errors and awkward phrasings (e.g., "This value is marked with superscript index *" or "among non-smoking women... expect for DPP-4"). A thorough language polish is recommended.

Author Response

Bizon and colleagues investigate the serum concentrations of incretin hormones (GIP, GLP-1) and their regulatory enzyme (DPP-4) in women with polycystic ovary syndrome (PCOS), examining their associations with metabolic parameters, sex hormone levels, and exposure to tobacco smoke. Through detailed subgroup analyses stratified by smoking status, insulin resistance (HOMA-IR), and body weight, the study reveals elevated levels of GIP and GLP-1 in PCOS patients, along with reduced DPP-4 levels in smokers. It suggests that incretins may play significant roles in PCOS pathophysiology and may serve as diagnostic or therapeutic targets. Despite the significance of the content, the novelty could be better emphasized by more clearly distinguishing this study’s contribution relative to prior work.

Response 1: Thank you for your thoughtful comment. We agree that incretin hormones may play a significant role in the pathophysiology of PCOS. While previous research has explores incretin dysregulation in PCOS, our study is, to our knowledge, the first to specifically investigate the influence of tobaccos smoke on incretin hormone levels and DPP-4 concentration in women with PCOS. This novel angle integrates endocrine and environmental factors, providing new insights into how lifestyle exposure like tobacco may interact with metabolic and hormonal pathways in PCOS. We revised the manuscript to more explicitly highlight this novel aspect in the Results and Discussion sections.

Another issues are shown below:

A) The design is generally sound, but the manuscript would benefit from clearer articulation of inclusion/exclusion criteria, recruitment procedures, and the rationale behind chosen cut-off points (e.g., HOMA-IR ≥2.0 for IR).

Response A: Thank you very much for your valuable suggestions.

We have added detailed information describing the study population in lines 355-368. The updated text is provided below:

The study was conducted among a cohort of 88 women admitted to the Gynecological Endocrinology Clinic at the Silesian Medical University in Katowice, Poland. The diagnosis of PCOS was based on the Rotterdam criteria (2003), requiring the presence of at least two of the following three features: ovulatory dysfunction, biochemical and/or clinical hyperandrogenism, and polycystic ovarian morphology [50]. Sixty women met the diagnostic criteria for PCOS. Among them, 37 were non-smokers and 23 were smokers.The control group comprised 28 non-smoking women who were initially reffered for suspected PCOS; however, hormonal and imaging evaluations excluded the diagnosis, and their hormonal profiles were witin normal limits. Participants with hyperprolactinemia, type 1 or type 2 diabetes, hypertension, Cushing’s syndrome, adrenal tumors, or alcohol abuse were excluded from both the study and control groups. More details decribing the PCOS cohort were published previously [51]. Ethical approval was obtained from the Bioethical Committee of Wroclaw Medical University (approval number 222/2024).

Regarding the HOMA-IR cut-off point:

In accordance with the European Group for the Study of Insulin Resistance, which recommends a threshold of >2.0 for diagnosing IR in young Caucasian women with polycystic ovary syndrome, we adopted this value to define IR in our study. The corresponding referenc has been added to the References section [53].

B) The cross-sectional nature of the study limits causal inference, yet some interpretations, particularly regarding smoking’s role in reducing DPP-4, lean toward causality. This should be tempered and discussed more critically in the discussion section.

Response B: In line with the Reviewer’s comments, we have revised the discussion to present a more cautious and balanced interpretation of the findings regarding  DPP-4 levels and smoking. All edits are highlighted in blue in the revised manuscript. Please find the revised paragraph added at the end of Discussion section and Conclusion:

While GIP and GLP-1 concentrations appear to be useful markers of glucose regulation in women with PCOS, exposure to tobacco smoke and elevated cotinine levels may act as confounding factors. Depending on the degree of exposure, individualized therapeutic strategies involving GLP-1 and/or GIP analogues or DPP-4 inhibitors may be warranted.

Conclusion

In summary, our findings indicate that GIP and GLP-1 concentrations are closely associated with metabolic and hormonal disturbances in women with PCOS. These associations are strongly influenced by IR and adiposity, while tobacco exposure appears to have a more substantial effect on DPP-4 levels. Given these results, incretin hormones hold promise both as biomarkers and therapeutic targets in PCOS. However, these findings are preliminary and require validation in larger, more diverse population. 

C) The control group appears younger and leaner than the PCOS group, which may confound the interpretation of metabolic and hormonal differences. A more detailed comparison or statistical control for these variables is advised.

Response C: We appreciate this observation. Below is a summary table comparing age and BMI between the PCOS and control groups:

Variable

Mean value

Median value

Min

Max

Upper quartile

Lower quartile

Standard deviation

PCOS

Age

(years)

25.50

24.50

20.00

35.00

22.00

28.00

3,83

Control

26.71

27.00

17.00

38.00

22.00

31.00

5.87

PCOS

BMI

(kg/m2)

26.13

23.93

17.07

41.95

20.76

30.86

6.74

Control

22.62

21.63

16.30

35.34

20.57

23.03

4.58

Although the PCOS group tend to be slightly older and have a higher BMI, statistical testing revealed no significant differences between the groups in either variable.

D) The manuscript would benefit from more mechanistic discussion regarding how smoking could alter DPP-4 expression or secretion, especially given the weak literature support and the lack of strong correlation between DPP-4 and incretin levels.

Response D: We agree that mechanistic insights are limited. To address this, we conducted an expanded literature search. Using the PubMed search terms: „Dipeptidyl peptidase 4 smoking women”, only two relevant studies were found (doi: 10.1111/jdi.13921; doi: 10.1016/j.maturitas.2019.05.003), neither of which directly addresses the relationship between DPP-4, smoking, and women.

Broadening the search to „DPP-4 and smoking” yielded more results.  However, even these did not identify  specific mechanisms linking tobacco smoke exposure to changes in DPP-4 levels. Most publications instead focused on smoking’s association with cardiovascular diseases, type 2 diabetes, or general health risks.

Therefore, we have clearly emphasized in the revised manuscript that our  findings are preliminary. Additional studies are needed to explore the mechanistic pathways and confirm these associations in larger cohort.

E) Figures 1–3 presenting ROC curves are informative, but resolution and figure legends need improvement. Table captions should also be made more descriptive for standalone comprehension.

Response E: We improved the resolutions and figure legends present in Figures 1-3, and corrected Table 5 and results descriptions. All modifications are highlighted in blue in the revised manuscript.

F) Some content in the discussion repeats results verbatim or revisits points multiple times (e.g., DPP-4 and smoking), which reduces narrative flow.

Response F: Thank you very much for your valuable comment. We have thoroughly revised and streamlined the entire Discussion section to improve its clarity, avoid redundancy, and enhance narrative flow. All modifications are highlighted in blue in the revised manuscript.

G) All abbreviations should be defined at first mention in both the abstract and the main text. For example, “IR” appears in the abstract without definition.

Response G: Thank you for pointing this out. We have carefully reviewed the abstract and the full manuscript and ensured that all abbreviations are clearly defined at their first mention.

Comments on the Quality of English Language

The manuscript is mostly well-written, though there are occasional grammatical errors and awkward phrasings (e.g., "This value is marked with superscript index *" or "among non-smoking women... expect for DPP-4"). A thorough language polish is recommended.

Response: We have carefully reviewed and improved the language throughout the entire manuscript to eliminate  awkward phrasing and grammatical errors.

Reviewer 2 Report

Comments and Suggestions for Authors

Reviewer Comments

  1. The statistical analysis should be provided below the tables.
  2. The results of this study require further interpretation and should be more thoroughly compared with findings from previous studies.
  3. In the Results section, the authors have only reported p-values, which alone do not provide sufficient insight into the clinical relevance of the observed statistical significance. To enhance the interpretation of the findings, I recommend including the corresponding confidence intervals alongside the p-values.
  4. How might your results impact clinical practice? Could genetic testing be a feasible and valuable tool for these patients? Additionally, is such testing economically sustainable?
  5. The results are very interesting but limited if they do not lead to improved outcomes. Please explain the primary and secondary prevention methods that can be applied for these patients.

Author Response

1. The statistical analysis should be provided below the tables.

Response 1: Thank you for this recommendation. We have now included the relevant statistical analyses directly below the tables to enhance transparency and clarity.

2. The results of this study require further interpretation and should be more thoroughly compared with findings from previous studies.

Response 2: Thank you for this important suggestion. We have significantly expanded and improved the Discussion section to include a more comprehensive comparison with previous studies and to offer deeper interpretation of our findings. All revisions are highlighted in blue. 

3. In the Results section, the authors have only reported p-values, which alone do not provide sufficient insight into the clinical relevance of the observed statistical significance. To enhance the interpretation of the findings, I recommend including the corresponding confidence intervals alongside the p-values.

Response 3: In response to your suggestion, we have added 95% confidence intervals alongside the p-values below Tables 3 and 4 to provide a clearer picture of the precision and clinical relevance of our results.

4. How might your results impact clinical practice? Could genetic testing be a feasible and valuable tool for these patients? Additionally, is such testing economically sustainable?

Response 4: Thank you for your insightful and constructive questions.

In accordance with your comments and suggestions of Reviewer 1, we revised Discussion section and Conclusion to emphasize that our findings are preliminary and require validation in a larger cohort.

We did not observe statistically significant associations between GIP, GLP-1, and DPP-4. However,  it was particularly interesting and unexpected, that smoking women with PCOS had nearly  two-fold lower serum DPP-4 concentrations. We also found a statistically significant negative correlation between cotinine concentration and DPP-4 levels (-0.41; p<0.001); though this correlation was not observed in other subgroups.

Although GIP and GLP-1 concentrations may be useful indicators of  glucose regulation in PCOS, exposure to tobacco smoke and/or elevated cotinine levels could disrupt this mechanism. Depending on the degree of exposure, personalized therapeutic approaches (GLP-1 and/or GIP analogues or DPP-4 inhibitors) might be considered.

Given the current findings, we believe it is premature to make  clinical or economical conclusions based on our preliminary findings. Further studies are required to explore the feasibility and cost-effectiveness of personalized treatments including the potential role of genetic testing. Although genetic analysis was not performer in this study, we agree that it would be a valuable direction for future research and help identify individual susceptibility to metabolic dysregulation in PCOS.       

5. The results are very interesting but limited if they do not lead to improved outcomes. Please explain the primary and secondary prevention methods that can be applied for these patients.

Response 5: Thank you for raising this critical point. Based on the clinical experience of Dr hab. Grzegorz Franik, Professor of Medical University of Silesia, we present below relevant primary and secondary prevention strategies for women with PCOS. Key elements of these recommendations have also been incorporated into Discussion section in lines 265-268.

Primary prevention aims to prevent the development of metabolic and cardiovascular complications in women with PCOS, especially in those with elevated incretin levels and early metabolic dysfunction potentially worsened by smoking. Key strategies include:

  • lifestyle modification (dietary intervention; physical activity - engage in at least 150 minutes per week of walk, moderate aerobic exercise and resistance training to enhance insulin sensitivity and reduce visceral fat; smoking cessation - as smoking is associated with increased oxidative stress and deceased DPP-4 concentration, might worsened insulin resistance
  • weight management – even 5-10% weight loss can significantly improve insulin resistance, hormonal balance, and ovulatory function)
  • patient education and monitoring

Secondary prevention focuses on halting progression and managing established metabolic disorders:

  • pharmacologic therapy (including metformin, GLP-1 Receptor Agonists, DPP-4 inhibitors, GIP/GLP-1 Co-Agonists
  • smoking cessation programs (essential to reduce cardiometabolic risk)
  • management of comorbidities (regular monitoring and management of type 2 diabetes, hypertension, dyslipidemia, non-alcoholic fatty liver disease)
  • reproductive health (addressing anovulation and menstrual irregularities with appropriate hormonal therapies or ovulation induction agents).

Round 2

Reviewer 1 Report

Comments and Suggestions for Authors

The authors provided a revised version of the manuscript with significant improvements.

Author Response

The authors provided a revised version of the manuscript with significant improvements.

Response: Thank you for your kindness and truly motivating feedback.